# Identifying multivariate disease trajectories and potential phenotypes of early knee osteoarthritis in the CHECK cohort

Sara Altamirano[1], Mylène P. Jansen[1]*, Daniel L. Oberski[2,3], Marinus J. C. Eijkemans[3], Simon C. Mastbergen[1], Floris P. J. G. Lafeber[1], Willem E. van Spil[1,4], Paco M. J. Welsing[1]

1 Department of Rheumatology & Clinical Immunology, University Medical Center Utrecht, Utrecht, The Netherlands, 2 Department of Methodology and Statistics, Faculty of Social and Behavioral Sciences, Utrecht University, Utrecht, the Netherlands, 3 Department of Data Science and Biostatistics, Julius Center, University Medical Center Utrecht, Utrecht, the Netherlands, 4 Department of Rheumatology, Dijklander Hospital, Hoorn, The Netherlands

* m.p.jansen-36@umcutrecht.nl

## Abstract

### Objective

To gain better understanding of osteoarthritis (OA) heterogeneity and its predictors for distinguishing OA phenotypes. This could provide the opportunity to tailor prevention and treatment strategies and thus improve care.

### Design

Ten year follow-up data from CHECK (1002 early-OA subjects with first general practitioner visit for complaints ≤6 months before inclusion) was used. Data were collected on WOMAC (pain, function, stiffness), quantitative radiographic tibiofemoral (TF) OA characteristics, and semi-quantitative radiographic patellofemoral (PF) OA characteristics. Using functional data analysis, distinctive sets of trajectories were identified for WOMAC, TF and PF characteristics, based on model fit and clinical interpretation. The probabilities of knee membership to each trajectory were used in hierarchical cluster analyses to derive knee OA phenotypes. The number and composition of potential phenotypes was selected again based on model fit (silhouette score) and clinical interpretation.

### Results

Five trajectories representing different constant levels or changing WOMAC scores were identified. For TF and PF OA, eight and six trajectories respectively were identified based on (changes in) joint space narrowing, osteophytes and sclerosis. Combining the probabilities of knees belonging to these different trajectories resulted in six clusters ('phenotypes') of knees with different degrees of functional (WOMAC) and radiographic (PF) parameters; TF parameters were found not to significantly contribute to clustering. Including baseline characteristics as well resulted in eight clusters of knees, dominated by sex, menopausal status and WOMAC scores, with only limited contribution of PF features.

**Data Availability Statement:** Restrictions on sharing the data for this study are imposed by the Institutional Review Board of the University

Medical Center Utrecht, Utrecht, The Netherlands. Data sharing is restricted because the dataset contains possible identifying information. All relevant data are available upon request by sending an email to the Rheumatology department of the UMC Utrecht (urrci@umcutrecht.nl). This is a non-author email address that allows for maintenance of long-term data accessibility. The full CHECK dataset, of which we used data for the current manuscript, is publicly available through the 'THEMATIC COLLECTION: CHECK (COHORT HIP & COHORT KNEE)', accessible through the DANS database (https://doi.org/10.17026/dans-252-qw2n).

**Funding:** This project was funded by ReumaNederland (project number 18-2-202). The funders had no role in study design, data collection and analysis, decision to publish, or preparation of the manuscript.

**Competing interests:** The authors have declared that no competing interests exist.

## Conclusions

Several stable and progressive trajectories of OA symptoms and radiographic features were identified, resulting in phenotypes with relatively independent symptomatic and radiographic features. Sex and menopausal status may be especially important when phenotyping knee OA patients, while radiographic features contributed less. Possible phenotypes were identified that, after validation, could aid personalized treatments and patients selection.

## Introduction

Osteoarthritis (OA) is the most common form of arthritis worldwide and the knee is among the most affected joints [1]. Its high prevalence is further increasing due to the aging population as well as increasingly widespread risk factors, especially obesity and a sedentary lifestyle [2]. It is estimated that more than 300 million people around the world currently suffer from OA and that the prevalence in the population aged $\geq$45 will increase from 26.6% in 2012 to 29.5% in 2032, partially due to increasing obesity [3, 4].

On average, people start experiencing OA symptoms around the age of 55 years and live 26 years with the condition [5]. OA can typically be described by a variety of symptoms and structural and biochemical joint changes in multiple tissues including bone, cartilage and synovium. The main symptoms of knee OA are pain, stiffness, and loss of function, leading to reduced mobility and quality of life. Conservative treatment options are focused on symptom reduction and are usually modestly effective at population level. Disease-modifying treatments with the ability to positively affect both the symptomatic and structural disease course are still not available [6].

As OA is hypothesized to be a heterogeneous disease that consists of different phenotypes, it is now believed that personalized diagnostics and treatment are required to optimize intervention [7]. It is plausible that disease heterogeneity is most perceptible and relevant in early-stage OA, as different etiologic processes may accumulate and coalesce over time to a final common pathway [8]. Moreover, as symptoms are generally more manageable and less advanced in early-stage OA, this is likely the most opportune moment to achieve modification of the disease course.

OA phenotypes have been defined as "*subtypes of OA that share distinct underlying pathobiological and pain mechanisms and their structural and functional consequences*" [9]. The optimal way to identify phenotypes of OA patients and their clinical value is still under investigation and is considered fundamental for the advancement of OA research [7]. Knowledge on potential subgroups of knee OA would allow further research into relevant etiological mechanisms and, in due course, provide insights for more effective prevention and treatment strategies. Such strategies are highly anticipated as they would help improve pain, functional disability, and quality of life for patients, as well as reduce societal burden and healthcare costs related to the disease.

In this study, we aimed to identify knee OA phenotypes using data from Cohort Hip and Cohort Knee (CHECK), consisting of 1002 individuals with symptoms and/or signs of early-stage knee and/or hip OA, having visited their general practitioner for the first time for these symptoms $\leq$6 months before inclusion. This large group of individuals was followed for 10 years and clinical data and radiographic data of both knees was collected, allowing analysis of longitudinal trajectories and interrelations of OA symptoms and radiographic features over time to identify phenotypes of knee OA.

## Methods

### Study population

CHECK, an initiative of the Dutch Arthritis Foundation, is a Dutch multi-center, 10-year prospective cohort study of 1002 individuals with symptoms and/or signs of early symptomatic OA of hip and/or knee, aged 45–65 years at the time of inclusion, and without a previous healthcare consultation or with a first consultation no longer than six months ago for these complaints to their primary care physician. Participants were evaluated clinically through regular physical examinations and questionnaires, radiographically through knee and hip radiographs, and biochemically through assessment of markers in plasma, serum, and urine samples [10]. Participants were examined and provided radiographs and samples at baseline (year 0) and, 2, 5, 8 and 10 years thereafter. For patients with more severe complaints at baseline (having 2 of pain, morning stiffness <30 min, crepitus and bony tenderness for the knee and/or 2 of pain, morning stiffness <60min pain on hip internal rotation or internal rotation <15˚; most patients), yearly questionnaire data was collected [11]. As a result, compared to these patients with yearly visits, a small group of patients with less severe complaints (9%) missed three or more visits and where excluded from all analyses. The CHECK study was approved by the medical ethics committees of all participating centers, and all participants gave their written informed consent before entering the study. Data collected in the CHECK study is publically available upon request [11, 12].

### Functional data analysis

To identify possible phenotypes, an unsupervised learning approach was used. Functional data analysis allows for division/clustering of subjects based on the course over time of multiple separate characteristics. For clinical signs and symptoms, we used the data of the WOMAC [13] subscales for pain, function, and stiffness in the knees. For structural signs, continuous measures of radiographic OA features were used for the tibiofemoral (TF) joint: joint space width (JSW), subchondral bone density, and osteophyte area, all measured medially and laterally using the Knee Images Digital Analysis (KIDA) method [14, 15]. Moreover, for the patellofemoral (PF) joint, semi-quantitative grades (0–3) of radiographic OA features were used: joint space narrowing (JSN), sclerosis and osteophytes, as scored on skyline radiographs [16]. For all features, data from all time points was used. We removed records of knees with missing data for three or more time points (of eleven possible time points for clinical features and five possible time points for radiographic features) and used linear interpolation to input remaining missing values. Baseline values of all parameters used for the functional data analysis and hierarchical cluster analysis were compared between included and removed records using Mann-Whitney U tests and Chi-square tests for continuous and categorical variables, respectively. Thereafter, to classify knees regarding these outcome trajectories in the course of OA, model-based clustering of functional data was performed on the included records using R packages *funHDDC* [17] and *fda* [18, 19]. This method aims to cluster the observed functional curves into $K$ homogeneous trajectory groups [20]. We selected the number of trajectory clusters ($K$) based on the Bayesian Information Criterion (BIC) [21] and evaluation of graphical representations of trajectory groups with different $K$'s by experts in the subject matter, also taking the number of individual knees in each trajectory group into account (without a formal minimum).

### Hierarchical cluster analysis

To define possibly relevant phenotypes, patients should be clustered based on both their characteristics (non-functional data) and their (clinical and structural) progression over time

(functional data), but this is not (directly) possible in a single analysis. To combine the functional data (expressed as probabilistic membership of a subject/knee to belong to a specific trajectory group found with the functional data analysis) and non-functional data (i.e. subject characteristics possibly relevant to define phenotypes (see below)) we used a hierarchical cluster analysis with Euclidean distance and Ward linkage method [22]. We used a stepwise approach to hierarchical cluster analysis starting with the trajectory groups (for WOMAC, TF and PF features) and then adding possibly relevant patient characteristics. The following characteristics were investigated based on possibly being related to experience of complaints and biomechanical or tissue processes: sex, menopausal status (if menopausal state was missing in females (22%) age > 50 was used), age, body mass index (BMI), biochemical markers in blood and urine, and pain-coping inventory (PCI) measures [23]. In explorative iterations we inspected clustering patterns using heatmaps created based on selected variables with the R packages *heatmaply* [24] and *dendextend* [25]. The specific data to include in the final model and number of phenotypes was again selected based on statistical consideration (average silhouette method (ASM) [26]), clinical considerations, interpretability and size of the clusters.

Differences in subjects' baseline characteristics between clusters were explored and tested using chi-square for categorical variables and Kruskal-Wallis Rank sum test for continuous variables. We followed reporting recommendations from the consensus-based framework for conducting and reporting osteoarthritis phenotype research [9].

## Results

### Functional data analysis

From the 1,002 subjects (2,004 knees) available in CHECK, 819 subjects (82%) and 1,788 knees (89%) had sufficient longitudinal data on all clinical and radiographic features for trajectory analysis (S1 Table). Baseline values did not differ between included and excluded knees (all p>0.05), except for TF medial JSW (median and interquartile range (IQR): included 4.66 (4.16–5.16); excluded 4.76 (4.26–5.26); p = 0.038) and PF JSN (included 0.00 (0.00–0.00); excluded 0.00 (0.00–0.00); p = 0.003; % JSN 0/1/2/3 for included 85.9/11.2/2.5/0.4, for excluded 85.8/8.8/5.1/0.3). During several iterations of data exploration no clear patterns could be obtained using all features in a combined analysis. Therefore, it was decided to perform separate analyses for 1) WOMAC pain, 2) WOMAC function, and 3) WOMAC stiffness, for 4) radiographic TF OA features and for 5) radiographic PF OA features, respectively. This was also deemed more appropriate because WOMAC scores were assessed at patient level as opposed to the radiographic features that were assessed at knee level, and because radiographic features were measured continuously for TF OA but discretely for PF OA. The finally derived trajectory groups for the different outcome sets are shown in Figs 1–3.

Trajectories for WOMAC scores were defined at patient level, for the TF OA and PF OA scores, analysis was performed and presented at the knee level as each knee in a patient may be differently affected by OA, and it was also considered possible that joints differed regarding their phenotype.

Regarding WOMAC pain, function and stiffness, five trajectories were discovered which could be named as (i) increasing, (ii) decreasing, (iii) high-stable, (iv) moderate-stable, and (v) low-stable (see Fig 1). Most patients were part of the moderate-stable (38–40%) or low-stable (23–39%) trajectories; the other trajectories contained 5–15% of patients.

For radiographic TF OA features, we identified eight trajectories. All trajectories showed decreasing medial JSW and increasing lateral JSW suggesting a population of primarily medial knee OA (although the most affected compartment was not determined). The derived TF OA trajectories were named as: (i) low osteophytes, moderate bone density, (ii) increasing bone

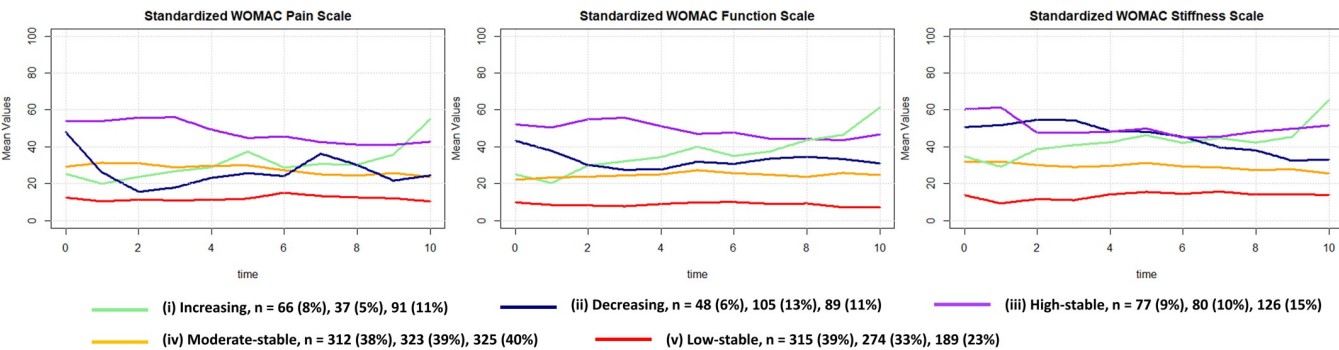

**Fig 1. Functional data analysis-based trajectories for WOMAC scores for pain, function, stiffness.** Average values within trajectory groups are shown.

density, (iii) low bone density, (iv) increasing lateral osteophytes, moderate bone density, (v) increasing osteophytes, low medial joint space width, increasing bone density, (vi) moderate-high bone density, (vii) high bone density, (viii) moderate-increasing bone density. Fig 2 shows the TF OA trajectories for medial and lateral TF OA features separately. The most pronounced changes occurred in osteophytes, although only for trajectories (iv) and (v), while JSW showed relatively little variation and change.

For radiographic PF OA characteristics, we found six trajectories and named them as: (i) low joint space narrowing, moderate osteophytes (ii) moderate-increasing OA features, (iii) low OA features, (iv) low joint space narrowing, low increasing osteophytes, (v) high-increasing OA features, (vi) high osteophytes (Fig 3). All three PF features showed relatively high change over time for some trajectories and variation between trajectories.

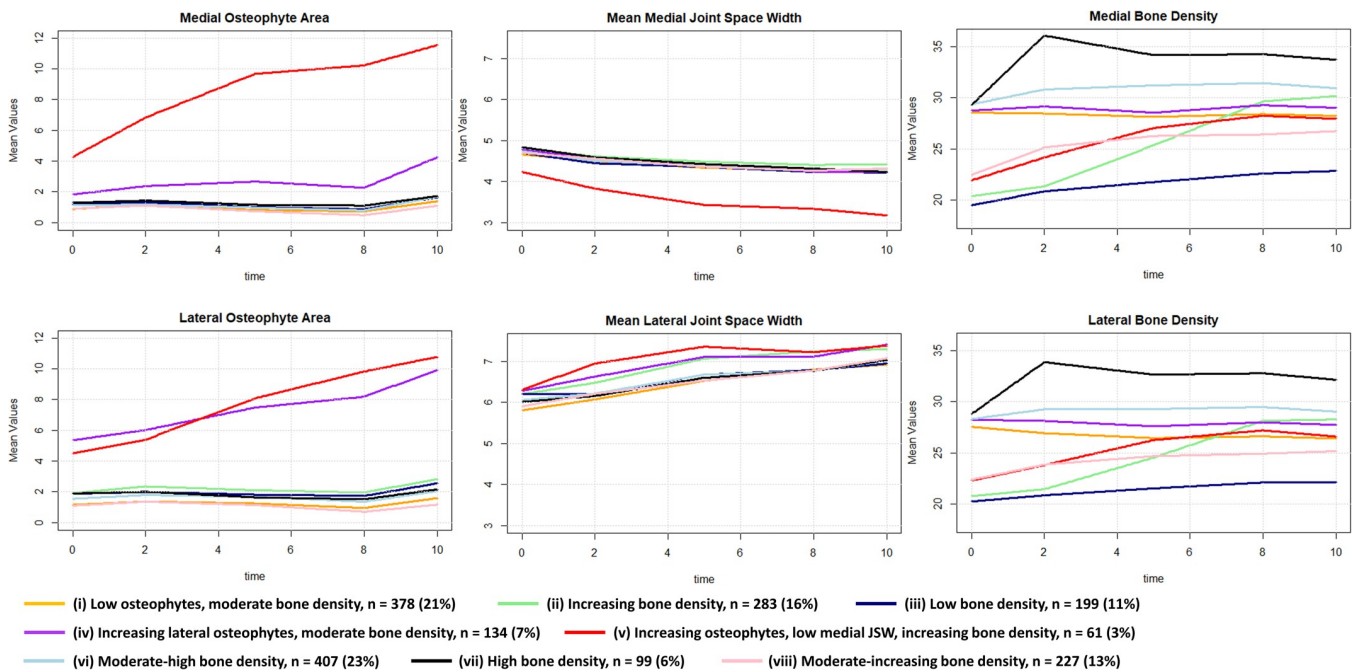

**Fig 2. Functional data analysis-based trajectories for quantitative tibiofemoral OA features: Osteophytes, joint space width, bone density.** Average values within trajectory groups are shown.

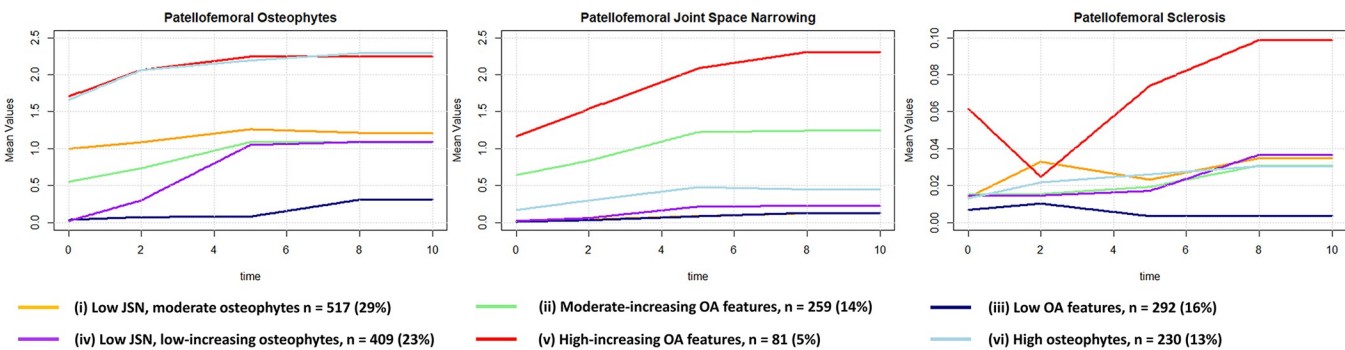

**Fig 3. Functional data analysis-based trajectories for OA patellofemoral features: Sclerosis, joint space narrowing, osteophytes.** Average values within trajectory groups are shown.

## Hierarchical Cluster Analysis (HCA)

Performing HCA with WOMAC pain, function and stiffness trajectories resulted in four clusters as the optimal solution that could be defined as low clinical impact, high clinical impact, increasing clinical impact, and decreasing clinical impact in line with the individual WOMAC trajectories. Subsequently, the PF OA characteristics were added. Resulting heatmaps and ASM suggested four clusters as statistically optimal, but PF OA damage was inconsistent within the clusters so after taking into account clinical considerations, a solution of six clusters was chosen as optimal (see below). Thereafter, we combined WOMAC and TF OA trajectories. However, inclusion of TF OA characteristics did not prove useful, since none of the cluster groups seemed to be driven by these TF OA characteristics, as determined graphically based on the heatmaps and changes in cluster division within the dendogram. The four groups resulting from clustering based on WOMAC and TF OA characteristics were very similar to the solution using WOMAC scores only, as 76% of knees were clustered in comparable groups. When combining WOMAC, PF OA characteristics, and TF OA characteristics, again TF OA did not add to the clustering solution.

To continue exploring potential phenotypes, the other selected patient characteristics (sex, menopausal status, age, BMI, biochemical markers (see S2 Table), and PCI) were added to the WOMAC and PF OA trajectories. Most of the added characteristics did not show any variability between derived clusters upon visual inspection of heatmaps. Only sex and menopause (categorized as men, premenopausal women, postmenopausal women) clearly differentiated between clusters, actually dominating the clusters. Therefore, we decided to remove the other baseline characteristics that mainly appeared to add noise.

Adding the TF OA trajectories again to the latter solution, or to a solution without the sex/menopause and including the other relevant characteristics, again confirmed that TF OA did not add to clustering solutions (results not shown).

Ultimately, two solutions were selected as clinically and statistically optimal. First, combining WOMAC pain, function, and stiffness with PF OA features which showed the following six clusters that were described as: (i) Low JSN, low WOMAC pain and function, (ii) Low WOMAC pain and function, (iii) high WOMAC pain and function, (iv) Moderate WOMAC features, low JSN, (v) Moderate WOMAC features, and (vi) Low increasing osteophytes, low JSN, moderate WOMAC features (see Fig 4). S2 Table describes patient characteristics for these final 'phenotype' clusters. Clusters differed statistically significantly on *smoking, hip endo- and exorotation active range of motion, lateral and medial bone density*. For example, the cluster with high WOMAC pain and functional complaints (vi) was less likely to smoke and

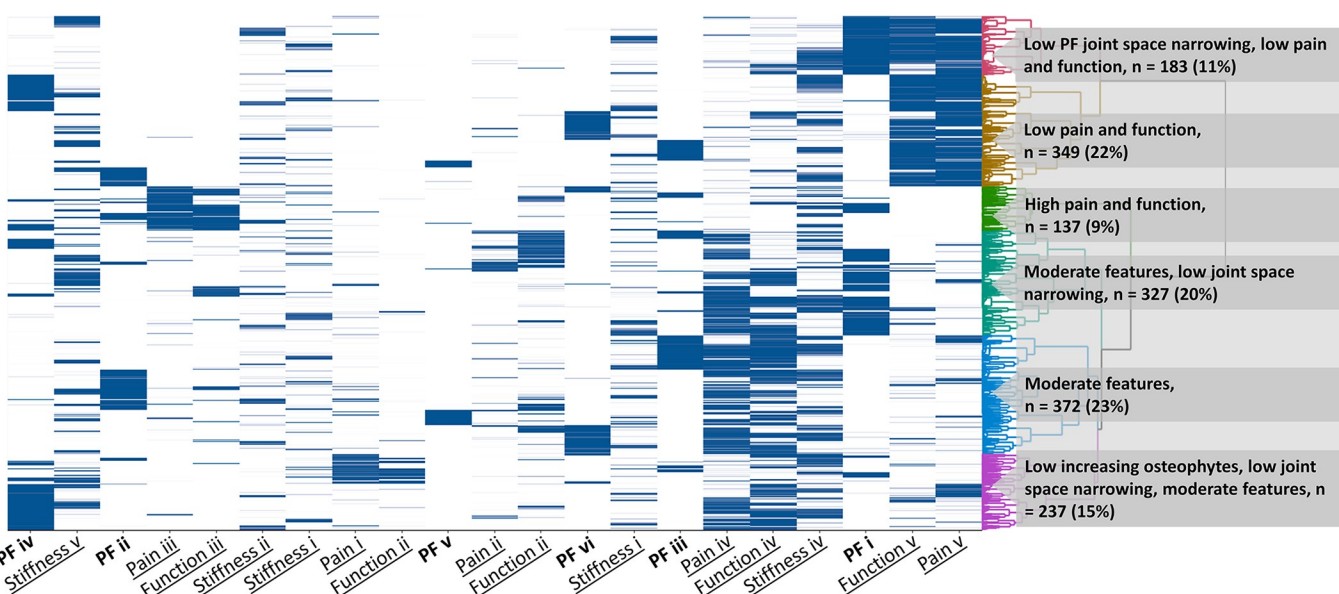

**Fig 4. Clustering based on WOMAC features (pain, function, stiffness) and patellofemoral (PF) osteoarthritis without sex.** The different WOMAC (underscore) and PF (bold) trajectories are indicated on the x-axis. WOMAC pain, function and stiffness trajectories are: (i) increasing, (ii) decreasing, (iii) high-stable, (iv) moderate-stable, and (v) low-stable. PF trajectories are: (i) low joint space narrowing, moderate osteophytes (ii) moderate-increasing OA features, (iii) low OA features, (iv) low joint space narrowing, low increasing osteophytes, (v) high-increasing OA features, (vi) high osteophytes.

had better hip ROM, while the cluster with moderate WOMAC and low JSN (iv) had somewhat higher TF bone density.

The other solution added sex and menopausal status as a feature which resulted in sex-dominated clusters. The resulting clusters were named: (i) Premenopausal females with low and moderate pain and function, (ii) Males with low pain and function, (iii) Males with moderate pain and function, (iv) Postmenopausal females with low pain and function, (v) Postmenopausal females with moderate pain and function and low joint space narrowing, (vi) Postmenopausal females with moderate pain and function and varying PF OA characteristics, (vii) Postmenopausal females with high pain and function, and (viii) Postmenopausal females with increasing WOMAC pain and function (Fig 5). These eight clusters differed statistically significantly on almost all characteristics without a clear trend, as shown in S3 Table.

## Discussion

In this study we identified trajectories of patient reported knee symptoms as well as tibiofemoral and patellofemoral radiographic characteristics of knee OA. Trajectories could be classified based on the absolute level over time of specific radiographic features (i.e. JSW, osteophytes, sclerosis) as well as their course over time. Some of these trajectories might be classified as progressive subtypes (i.e. for TF OA increasing lateral and medial osteophytes, low medial JSW, increasing bone density, and for PF OA increasing/high sclerosis and JSN, and high osteophytes) or non-progressive (i.e. for TF low osteophytes, stable bone density, or for PF OA, low sclerosis, JSN, and osteophytes). Likewise, for symptoms, groups with low, moderate and high levels of symptoms over time as well as increasing and decreasing trends over time were identified.

As our primary aim was to identify knee OA phenotypes based on radiographic features as well as symptoms and possible etiologically relevant patient characteristics, we combined the trajectory groupings for each of these features in a cluster analysis. Results indicated that these

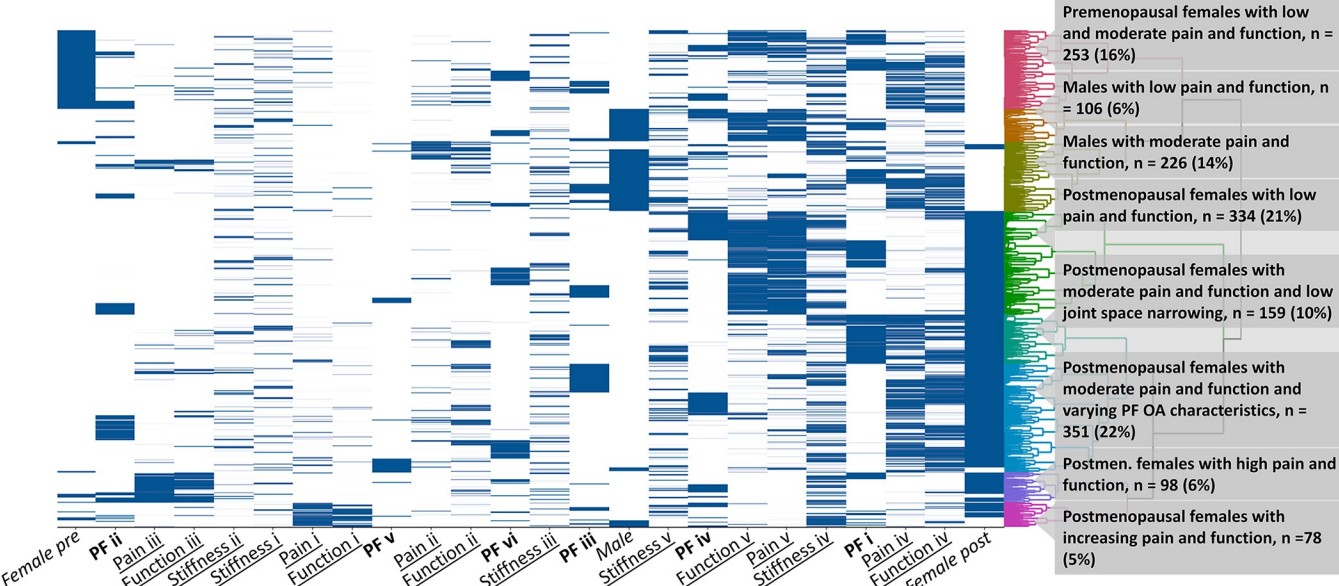

**Fig 5. Clustering based on WOMAC features (pain, function, stiffness) and patellofemoral (PF) osteoarthritis with sex.** Sex is split into male, female pre-menopause, female post-menopause. The different WOMAC (underscore) and PF (bold) trajectories and sex (italics) are indicated on the x-axis. WOMAC pain, function and stiffness trajectories are: (i) increasing, (ii) decreasing, (iii) high-stable, (iv) moderate-stable, and (v) low-stable. PF trajectories are: (i) low joint space narrowing, moderate osteophytes (ii) moderate-increasing OA features, (iii) low OA features, (iv) low joint space narrowing, low increasing osteophytes, (v) high-increasing OA features, (vi) high osteophytes.

features (and their course over time) are relatively independent. Specifically, the radiographic TF OA characteristics did not seem to add to subgrouping/phenotyping knees and radiographic PF OA features did contribute only to a limited extent. This is, however, in line with previous findings [27–29] suggesting that TF OA features are not strongly related to clinical features. The fact that PF OA was (somewhat) more related to complaints may indicate that PF OA is typically an early sign of the development of knee OA and the TF OA characteristics follow later, and thus may not influence phenotypes in this cohort of early OA. This has been suggested before [30]. Alternatively, the absence of TF features in the clustering might be the result of the continuous measurements used for TF features, sensitive to inter-subject variation, instead of semi-quantitative measurement of PF features. While this does directly not matter for the clustering, since only trajectory probabilities were used and no measurement values, it may have influenced the identified trajectories. Additionally, general patient characteristics were of limited help in subgrouping. Only sex and menopausal status seemed to influence subgrouping, which resulted in an almost complete separation between males and pre- and post-menopausal women, with further subgrouping based on reported knee complaints. Although, the analyses seemed to be somewhat dominated by sex and menopausal state, it may also suggest that taking these factors into account in defining phenotypes may indeed be important from a pathobiological view. Also, it is not clear whether importance of sex in the phenotyping is caused by biological differences (e.g. hormones, which are influenced by menopause as well) or social differences, or both.

There is a chance findings do not point to different phenotypes, but to different disease stages. However, there is no obvious chronologic line in the clusters (i.e. you cannot simply sort them from less to more advanced OA phase). Also, all patients have had their first complaints within 6 months of the first visit, meaning they are clinically all in the same phase of the disease at baseline, i.e. have the same starting point. Although this starting point is only

based on clinical outcome, and not on radiographic features, it is very relevant in clinical practice. Further, the baseline KL grade and most tibiofemoral parameters were comparable and not significantly different between the found clusters (S2 Table), so they likely did not start at very different structural stages either. Even if we derived trajectories represent different disease phases, groups would (at least partly) represent patients that progress more or less quickly through different phases, which may be regarded in itself a phenotype.

While some of our results were anticipated, we expected more trajectories showing discordant courses between radiographic features and symptoms (e.g. a subgroup of patients with severe symptoms and only limited radiographic signs, reflecting central pain sensitization). Instead, radiographic features and symptoms mainly showed concordant courses and only the PF features 'low-increasing osteophytes, low JSN' and the clusters containing 'moderate WOMAC features, low JSN' could be considered somewhat discordant. On the other hand, the different clusters based on WOMAC severity did not show significant differences in tibiofemoral features (e.g. JSW, osteophytes, KL grade), as shown in S2 Table, indicating that there is some discordance between symptoms and radiographic TF features, but this is not expressed in the clustering since TF features did not contribute. Also, we expected to see a distinction between lateral and medial OA, potentially with different results for patients with predominantly lateral or medial OA, but this was not the case. As medial TF knee OA is generally more prevalent, the number of patients with predominant lateral TF knee OA might have been too low to be represented in separate trajectories. Since medial *vs* lateral OA was not officially determined, it is unknown how many participants had medial or lateral OA. Increasing the number of trajectories and/or clusters might have resulted in finding a lateral OA group, but even with defining up to 10 clusters no clear lateral OA group emerged, indicating that lateral OA is not an important subgroup of early OA.

Other phenotype analyses have been performed in OA cohorts before [31]. Two studies used data from the CHECK cohort as well and identified distinct trajectories in the Numeric (Pain) Rating Scale (NRS) over five years. While one showed similar results as we did for the ten-year WOMAC trajectories [32], the other showed trajectories based only on the level of pain and not the change in pain over time [33]. Both studies, as well as another CHECK study identifying WOMAC function over five years, showed significant differences between trajectory subgroups in BMI, education level, and comorbidity [34]. In our clusters based on WOMAC and PF features, we did not find significant differences in these characteristics (all p>0.07; S2 Table). However, when comparing patients from different WOMAC trajectory subgroups, we found similar differences for the number of comorbidities (p = 0.001) but not BMI or education level (both p>0.14). Between clusters that were also based on sex and menopausal status all these three characteristics differed significantly, but these differences may be driven by the sex and menopausal status differences. The fact that we did not find differences between groups for BMI or education level could be because we did not compare characteristics between trajectories as the previous studies did, but between clusters based on more than only clinical outcome, and possibly because of the longer follow-up of ten instead of five years. Other multivariate clustering studies, using different patient cohorts, have been performed as well, although clustering was based on baseline data and not on changes over time in those studies, and PF characteristics were not taken into account [35, 36]. Like in our analyses, those clusters showed differences in OA structural damage or pain, but unlike in our study, they differed significantly in BMI and, in one study, results showed separate clusters for medial or lateral OA. The fact that previous studies found significant differences in BMI and we did not might be because previous studies clustered based only on pain, and BMI is known to be linked to musculoskeletal pain, while we looked at other factors as well.

This study naturally had limitations. Some data was missing (4%, 13% and 6% of visits for WOMAC, TF and PF parameters, respectively), which was imputed using linear interpolation to enable functional data analysis. While we think the impact of this missing data was minimal, it may still have influenced subgrouping results. Also, excluding patients with ≥3 visits means patients with less severe complaints and therefore without yearly visits were excluded. This might have caused selection bias, although the baseline complaints (clinical outcome) were not different between included and excluded patients, and looking at the WOMAC trajectories, the population clearly still included patients with less severe complaints. JSW/JSN did show a small but significant difference between included and excluded patients, with the excluded patients showing a somewhat less affected JS. Furthermore, the fact that only radiographs were available for evaluation of structural TF and PF parameters was a major limitation. Using other imaging modalities such as MRI or CT may have allowed for analysis of additional relevant structural parameters (e.g. bone marrow lesions) or for better evaluation of the current parameters (e.g. directly evaluating cartilage thickness). Further, while the results were based on statistical and expert-knowledge based considerations, final results are sensitive to the expert knowledge of the researchers as well as the data source (as is always the case).

Because of the nature of the CHECK cohort, patients with predominantly hip OA as well as patients without (knee) OA could have been included. As such, while most patients likely had early knee OA, patients without knee OA were included in these results as well, which might have influenced and be represented in the trajectories and clustering with few progression and complaints. Also, while the WOMAC questionnaire is specific to the knee, having hip OA might influence knee function as well, as could the contralateral knee and hip, since the WOMAC was available only on patient level and not on joint level. An important next step in this research would be to validate our results in another cohort with early-stage knee OA or another cohort with similar measurements like the OsteoArthritis Initiative (OAI). Moreover, for some of the identified trajectories/phenotypes, it may be of value to predict if a patient will likely belong to that group in an early phase of the disease. For example, more progressive groups (e.g. the 'increasing osteophytes, low medial JSW, increasing bone density' trajectory or 'postmenopausal females with increasing pain and function' phenotype) may be most likely to be destined for joint replacement, although this did not clearly seem to be the case during the 10-year follow-up of CHECK (S2 and S3 Tables). This would have to be evaluated with prediction algorithms and validation in a future study with long follow-up.

In conclusion, based on our results, patient trajectories in pain and radiographic features seem to be largely independent, and sex and menopausal status need to be considered when phenotyping knee OA patients. While further validation is required as a next step, interrelations of OA symptoms and radiographic features were analyzed to identify phenotypes in a large group of early knee OA patients, which in the future could be used for personalized treatments and patient selection.

## Supporting information

**S1 Table. Final list of variables included in the functional analysis.** *We removed records of knees with missing data for three or more time points and used linear interpolation to input remaining missing values.
(DOCX)

**S2 Table. Descriptive statistics of combined scenario with WOMAC and PF OA features.** ROM: range of motion; CTX-I: C-terminal telopeptide of collagen I; CTX-II: C-terminal telopeptide of type II collagen; C1,2C: collagen of types I and II; COMP: cartilage oligomeric matrix protein; PIIANP: collagen N-propeptide of type IIA; CS846: chondroitin sulphate 846;

NTX-I: N-terminal telopeptide of collagen I; OC: osteocalcin; PINP: aminoterminal propeptide of type I procollagen; HA: hyaluronic acid; PIIIANP: N-terminal propeptide of type III procollagen; hsCRP: high-sensitivity C-reactive protein; BSE: erythrocyte sedimentation rate. * P-values calculated with chi-square test. **for missing values, mean of all SOS items was taken (n = 6). Median and interquartile range (Q1-Q3) given for continuous variables unless otherwise indicated.

(DOCX)

**S3 Table. Descriptive statistics of combined scenario with WOMAC, PF OA features and sex.** ROM: range of motion; CTX-I: C-terminal telopeptide of collagen I; CTX-II: C-terminal telopeptide of type II collagen; C1,2C: collagen of types I and II; COMP: cartilage oligomeric matrix protein; PIIANP: collagen N-propeptide of type IIA; CS846: chondroitin sulphate 846; NTX-I: N-terminal telopeptide of collagen I; OC: osteocalcin; PINP: aminoterminal propeptide of type I procollagen; HA: hyaluronic acid; PIIIANP: N-terminal propeptide of type III procollagen; hsCRP: high-sensitivity C-reactive protein; BSE: erythrocyte sedimentation rate. * P-values calculated with chi-square test.

(DOCX)

## Author Contributions

**Conceptualization:** Willem E. van Spil, Paco M. J. Welsing.

**Data curation:** Sara Altamirano, Mylène P. Jansen.

**Formal analysis:** Sara Altamirano, Mylène P. Jansen, Daniel L. Oberski, Marinus J. C. Eijkemans, Simon C. Mastbergen, Floris P. J. G. Lafeber, Willem E. van Spil, Paco M. J. Welsing.

**Funding acquisition:** Paco M. J. Welsing.

**Investigation:** Marinus J. C. Eijkemans, Simon C. Mastbergen, Floris P. J. G. Lafeber, Willem E. van Spil.

**Project administration:** Sara Altamirano, Mylène P. Jansen, Paco M. J. Welsing.

**Resources:** Paco M. J. Welsing.

**Supervision:** Daniel L. Oberski, Marinus J. C. Eijkemans, Simon C. Mastbergen, Floris P. J. G. Lafeber, Willem E. van Spil, Paco M. J. Welsing.

**Visualization:** Sara Altamirano, Daniel L. Oberski.

**Writing – original draft:** Sara Altamirano, Mylène P. Jansen.

**Writing – review & editing:** Daniel L. Oberski, Marinus J. C. Eijkemans, Simon C. Mastbergen, Floris P. J. G. Lafeber, Willem E. van Spil, Paco M. J. Welsing.

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
