## [Decision Letter · Decision Letter 0]

19 Dec 2022

PONE-D-22-19232Identifying multivariate disease trajectories and potential phenotypes of early knee osteoarthritis in the CHECK cohortPLOS ONE

Dear Dr. Jansen,

Thank you for submitting your manuscript to PLOS ONE. After careful consideration, we feel that it has merit but does not fully meet PLOS ONE’s publication criteria as it currently stands. Therefore, we invite you to submit a revised version of the manuscript that addresses the points raised during the review process.

ACADEMIC EDITOR: Revisions are required for this manuscript to improve clarity and details on the analysis and methods. 

We look forward to receiving your revised manuscript.

Kind regards,

Aqeel M Alenazi

Academic Editor

PLOS ONE

Journal Requirements:

2. In the Methods section of your revised manuscript, please include the full name of the institutional review board or ethics committee that approved the protocol, the approval or permit number that was issued, and the date that approval was granted.

Additional Editor Comments (if provided):

Revisions are required for this manuscript to improve clarity and details on the analysis and methods.

Reviewers' comments:

Reviewer's Responses to Questions

**Comments to the Author**

1. Is the manuscript technically sound, and do the data support the conclusions?

Reviewer #1: Yes

Reviewer #2: Yes

2. Has the statistical analysis been performed appropriately and rigorously? 

Reviewer #1: I Don't Know

Reviewer #2: I Don't Know

3. Have the authors made all data underlying the findings in their manuscript fully available?

Reviewer #1: No

Reviewer #2: Yes

4. Is the manuscript presented in an intelligible fashion and written in standard English?

Reviewer #1: Yes

Reviewer #2: Yes

5. Review Comments to the Author

Reviewer #1: This is an interesting and complex manuscript about describing the longitudinal progression of people with knee OA. My understanding of trajectories and clusters analyses are that they are descriptive analyses, retrospectively classifying people into groups.

My main comment is that I am unable to provide any meaningful review of the statistical analysis and the decision-making process. My knowledge of these techniques is too limited. I am also not certain that others could replicate the analysis process of this paper. I think there were multiple, iterative analyses which lead to results-driven decisions about the next stage. I believe that these investigator decisions are likely to be valid, but that possible limitations of this approach should be discussed. Is this liable to something similar to "over-fitting", or the usual limitations of post-hoc analysis?

I believe that all data from all time points were used in this paper. However, I didn't find it easy to confirm this opinion when I read the methods. Maybe make this clearer?

Surely the data collection method will have a strong influence on the final analyses? So the unusual decision to measure different "severities" at different rates needs a discussion. The authors mention that there was a lack of discordant findings in their clusterings. Could this have been influenced by the data collection frequencies?

I tried to check the trajectories and I think they look ok. I couldn't see any issues with their names/descriptions.

I think figures 4 and 5 could benefit from detailed long legends. I am not certain what the PFi, Pain iv etc x-axis labels describe.

In the supplements, the authors report different clusters based upon WOMAC severity, and they often have similar radiographic findings. This might be evidence of the discordance that the authors were looking for in their clusters? Is it surprising how often KL grade was similar across many clusters (supplement tables)?

In general I found the Discussion to be conservative. The impact of this paper will probably be due to classification of people into different outcomes in their future. Do the clusters/trajectories yield something that appears to be clinically-meaningful? Which groups of people seem most likely to be destined for joint replacement etc? One major finding was the separation of males from pre- and post-menapase females; did the final clinical outcomes of these groups differ in a meaningful way? Also, if the authors choose to increase their sex-based discussion, then a greater discussion of possible sex/gender-related confounding should be included.

Multiple clusters had relative small percentages of people in them. Is the failure to identify a lateral OA cluster a serious limitation?

Reviewer #2: Major comments:

As this is a study in a supposedly “early” OA cohort, it is a great pity that the characterization at a structural level was limited to radiography, as using several MRI features (cartilage and meniscus damage, meniscal extrusion, bone marrow lesions, synovitis) would have been much more informative, with radiography missing all these tissue specific pathologies Even JSW (and JSN) in radiography has been shown to not be specific to cartilage thickness, but is confounded by meniscus extrusion. The authors should specify this as a major limitation of the study.

This being said, the current analysis is of course still very useful, since MRI is not currently used in the clinical routine of managing knee OA.

Yet, I think the authors need to provide more evidence that they are examining really phenotypes, and not just different stages of the disease. There appear to be patients with and without radiographic signs of TF and PF disease, and exhibiting different grades. How can the authors ensure that entering these variables, they are not just differentiating stages of the same disease (same path of progression), rather than different phenotypes (different paths). Similar considerations apply to the WOMAC scores.

Minor comments:

ABSTRACT

Design: Please give a short hint (maybe in brackets) how early OA is defined.

Results:

How is the sentence: Combining….and radiographic (PF) parameters….

It is unclear from my perspective whether the six clusters are only based on PF radiographic OA, and why TF radiographic parameters are not included.

When you say “gender”, I think this should be termed “sex”, since you are talking about biological and not social attributes. This comment also applies to the main text.

When including baseline parameters, does none of the radiographic features contributes to any of the clusters ?

Conclusions:

Although several…., they were shown…. Why are the first and second part of the sentence contradictions. The sentence is pretty general and could be a bit more specific.

Are really “gender” and “menopausal status” the main components making up 6-8 clusters (including men?), when phenotyping knee OA patients. From this sentence it appears that functional and radiographic (PF and TF) make not contribution. If this is so, please state this in the conclusion. Again, the conclusion should, in my view, be more concrete and more specific.

INTRODUCTION

P9, line 50: Please omit “already”

P9, line 56: What is meant by “biochemical joint changes”. Please name the tissue you are referring to.

P9, line 59/60: … treatment….are available: Either “treatments” or “is available”

P9, line 61: I suggest to say: disease “that consists of different phenotypes”

P9, line 64: suggest to add: “coalesce over time to a final common pathway. (delete “in patients”)

P10, line76: Please state in the introduction how you define “early” OA, as this term is very fashionable and it appears every researcher understands something different when using this term.

P 10, line 78: Please omit “for the first time”

METHODS

P10, line 84. As this study appears to focus on knee OA, but some patients also have hip OA, did you consider to include hip OA in your analysis and phenotypic characterization, as lower limb function may be impacted by the presence / absence of hip OA as well as knee OA.

P10, lines 90/91: How can the patients have “severe complaints at baseline (most patients”, if this is a cohort of “early” disease?

P11, lines 95/96: Did you use WOMAC scores from one knee only (which one), or from both knees, as did the OAI. How did you account for contralateral knee (and hip) functional status in your analyses?

P 11, line 98: Why did you use an absolute measure of JSW that varies substantially between patients of different sex and/or different body height, instead of using the common JSN grading (comparing the JSW with the less affected compartment and/or contralateral knee), as a real measure of pathology.

P 11, line 98: Why did you use osteophyte “area” rather than the common semi-quantitative grading system?

P 11, line 99/100: Why did you use a quantitative system for the TF joint, and a semi-quantitative one for the PF joint? As you appear to find a greater contribution of the PF than TF joint to the clusters, may this be due to PF radiographic grades being used semi-quantitatively (as a sign of pathology), whereas the TF joint is evaluated quantitatively, where absolute measures are not really a sign of radiographic pathology, but are strongly confounded by inter-subject variation.

P 11, line 106: Thereafter…. Is this then referring only to the group included. This is a bit confusing.

P 11, line 111: domain “experts”. Is this word used correctly here. May be due to my unfamiliarity with this particular method.

P 12, lines 115-117: Can you give the non-specialist reader a sense why the non-functional data is possibly relevant to define phenotypes (and not the functional analysis)?

P12, lines 120-123: It appears that the PF and TP radiographic features were not included at this level; why not?

P12, lines 115 ff: Could you try to give the reader a sense of what these various methods do conceptionally, as not every reader may be familiar with theses statistical methods.

RESULTS

P13, lines 138-141: Why do you think specifically the TF JSW and PF JSN differ between included and excluded participants, but none of the other parameters?

P13, lines 141 ff: it was decided…. based on which criteria

P13, lines 149 ff: Why was the WOMAC score not determined at the knee level (as was done in OAI). Which knee was selected for the analysis regarding WOMAC in each patient? How do the authors expect the contralateral knee to confound the scores?

P13, lines 160 ff: How did you define medial knee OA in the absence of radiographic JSN scores?

P14, line 184: Adding WOMAC…. Adding to what ?

P15, lines 190 ff: I wonder whether the lack of relevance of TF OA is due to the lack of use of JSN scores, as JSW is not a measure of pathology in a cross sectional context?

Can you provide results on what the JSW change was over 10 years in the medial and in the lateral compartment, respectively?

P15, line 198 : Please do not say “unfortunate”, the presentation of results should be neural

P15, lines 200 ff: When sex and menopause were added (how did you classify men in this context), did really all the WOMAC and radiographic PF OA data became irrelevant (Therefore we decided to remove the other characteristics……)

P15, lines 206ff: In my view, these phenotypes appear to represent different stages of the disease.

P 15/16: The listing of the different clusters in the text is a bit difficult to follow. Maybe these could be put into Tables, cross-tabling the different features. This will also display visually which combinations were not in the set of selected phenotypes.

DISCUSSION:

P 16, lines 238 ff.: It would be beneficial if the structure of presentation of results could differ more clearly between cross sectional and longitudinal (change) distributions of the various features.

P 17: When talking about TF and PF OA here, please add “radiographic” to the description.

P 18, lines 268/269: How many of your participants had medial vs. lateral TF radiographic OA?

P 19, line 291: Why do the authors think the results are so discordant between studies, given that they have examined a relatively large cohort.

P 19, line 293: Instead or “some missing data was present”, please say “ some data was missing”.

P 19, lines 308 ff: Shouldn’t the Osteoarthritis Initiative (OAI) data be used for validation?

P 20, conclusions: There is more text about what was done than about what was found. Please focus more on concrete and specific findings in the conclusion section (also in the abstract)

P 20, conclusions: Please do not state “for the first time”, since the study is one in line of such analyses, and of course every study has its peculiarities, but these to not necessarily qualify for stating “for the first time”, if similar approaches have been taken before.

6. PLOS authors have the option to publish the peer review history of their article (what does this mean?). If published, this will include your full peer review and any attached files.

Reviewer #1: **Yes: **Daniel McWilliams

Reviewer #2: No

---

## [Author Response · Author response to Decision Letter 0]

26 Jan 2023

We thank the reviewers for the helpful comments. We addressed these comments and feel the manuscript has improved. Please find an itemized response to all questions of the reviewers below. Hopefully we answered all questions and addressed all comments to the intention and satisfaction of the reviewers.

Journal requirements

- Journal Comment: 

Please ensure that your manuscript meets PLOS ONE's style requirements, including those for file naming. The PLOS ONE style templates can be found at https://journals.plos.org/plosone/s/file?id=wjVg/PLOSOne_formatting_sample_main_body.pdf and https://journals.plos.org/plosone/s/file?id=ba62/PLOSOne_formatting_sample_title_authors_affiliations.pdf

- Author response:

The manuscript’s style and file naming have been updated to meet PLOS ONE’s style requirements (this has been done without tracked changes, for readability). 

- Journal comment:

In the Methods section of your revised manuscript, please include the full name of the institutional review board or ethics committee that approved the protocol, the approval or permit number that was issued, and the date that approval was granted.

- Author response:

For this study, we used publically available data originally collected in the CHECK cohort. As such, we do not have a name and number for the ethical approval that was granted. However, we have added the following to the Methods section of the revised manuscript [line 101-103], with references to the original study and the data: “The CHECK study was approved by the medical ethics committees of all participating centers, and all participants gave their written informed consent before entering the study. Data collected in the CHECK study is publicly available upon request.[11,12]”

- Journal comment:

Please include captions for your Supporting Information files at the end of your manuscript, and update any in-text citations to match accordingly. Please see our Supporting Information guidelines for more information: http://journals.plos.org/plosone/s/supporting-information.

- Author response:

Supporting Information file names and captions have been updated as required (without tracked changes, for readability).

Reviewer 1

- Reviewer comment: 

This is an interesting and complex manuscript about describing the longitudinal progression of people with knee OA. My understanding of trajectories and clusters analyses are that they are descriptive analyses, retrospectively classifying people into groups. My main comment is that I am unable to provide any meaningful review of the statistical analysis and the decision-making process. My knowledge of these techniques is too limited. I am also not certain that others could replicate the analysis process of this paper. I think there were multiple, iterative analyses which lead to results-driven decisions about the next stage. I believe that these investigator decisions are likely to be valid, but that possible limitations of this approach should be discussed. Is this liable to something similar to "over-fitting", or the usual limitations of post-hoc analysis?

- Author response: Indeed we use trajectory and cluster analysis to identify possible relevant subgroups in progression of knee OA. It is important to realize that we use these statistical techniques as a tool to enable clinical evaluation and definition of (clinically) relevant subgroups. The analysis is thus merely used for identifying and illustrating possible subgroups for clinical validation. This validation is to our opinion key and the advanced analyses helps to perform this tasks more sensitively. Overfitting is not an important issue here to our opinion, because results are a combination of statistical as well as expert knowledge-based feasibility. This may actually be better generalizable than either the statistically most feasible or expert knowledge-based solutions only. However, final results are, of course, sensitive to expert knowledge of the researchers as well as the data source (as is always the case). This has been added to the limitations in the Discussion section of the revised manuscript [line 354-356].

- Reviewer comment:

I believe that all data from all time points were used in this paper. However, I didn't find it easy to confirm this opinion when I read the methods. Maybe make this clearer?

- Author response:

Indeed data from all time points was used. We have clarified this in the Methods section of the revised manuscript [line 114].

- Reviewer comment:

Surely the data collection method will have a strong influence on the final analyses? So the unusual decision to measure different "severities" at different rates needs a discussion. The authors mention that there was a lack of discordant findings in their clusterings. Could this have been influenced by the data collection frequencies?

- Author response:

Unfortunately the decision to measure different severities at different rates was already taken, as we used previously collected data from the CHECK cohort. However, the lack of discordant findings in our clustering was likely not the result of the difference in data collection frequencies, since patients with less frequent visits (because of less severe complaints) missed ≥3 visits and were therefore excluded from all analyses in this manuscript. As such our population consisted of early OA with clinically relevant complaints. This is discussed in the Discussion section [line 345-350 of the revised manuscript with tracked changes].

- Reviewer comment:

I tried to check the trajectories and I think they look ok. I couldn't see any issues with their names/descriptions.

- Author response:

We are happy to hear that.

- Reviewer comment:

I think figures 4 and 5 could benefit from detailed long legends. I am not certain what the PFi, Pain iv etc x-axis labels describe.

- Author response:

These labels describe the different WOMAC pain, function, stiffness trajectories and patellofemoral (PF) trajectories resulting from the functional data analysis. We should have made this more clear in the legends of Fig4 and Fig5, and have now done so in the revised manuscript [line 236-240 and 253-257].

- Reviewer comment:

In the supplements, the authors report different clusters based upon WOMAC severity, and they often have similar radiographic findings. This might be evidence of the discordance that the authors were looking for in their clusters? Is it surprising how often KL grade was similar across many clusters (supplement tables)?

- Author response:

This is a very good point and indeed is indeed evidence that there is some discordance between radiographic (TF) features and symptoms, it just is not expressed in the clustering since TF features did not contribute to that. We have added this consideration in the Discussion section of the revised manuscript [line 307-310].

- Reviewer comment:

In general I found the Discussion to be conservative. The impact of this paper will probably be due to classification of people into different outcomes in their future. Do the clusters/trajectories yield something that appears to be clinically-meaningful? Which groups of people seem most likely to be destined for joint replacement etc? 

- Author response:

While development of prediction algorithms and validation is outside of the scope of the current paper, it is indeed important for confirming the clinical relevance of the found clusters/trajectories and we should indeed discuss this in the current paper. As mentioned by the reviewer, it is relevant to discuss that especially the more progressive groups (e.g. the ‘increasing osteophytes, low medial JSW, increasing bone density’ trajectory or ‘postmenopausal females with increasing pain and function’ phenotype) may be most likely to be destined for joint replacement, although this did not clearly seem to be the case during the 10-year follow-up of CHECK (Table S2/S3). We have now included this in the Discussion section of the revised manuscript [line 366-371].

- One major finding was the separation of males from pre- and post-menapase females; did the final clinical outcomes of these groups differ in a meaningful way? Also, if the authors choose to increase their sex-based discussion, then a greater discussion of possible sex/gender-related confounding should be included.

- Author response:

Especially progression of clinical features (i.e. WOMAC pain) was quite different between males and –pre- and post-menopausal females, which is why the phenotypes (Fig 5) are so clearly separated between sexes. It is not clear if the differences between sexes and menopausal status is because of biological differences (e.g. hormones, which are influenced by menopause as well) or social differences, or both. We have added this consideration to the Discussion section of the revised manuscript [line 288-290].

- Reviewer comment:

Multiple clusters had relative small percentages of people in them. Is the failure to identify a lateral OA cluster a serious limitation?

- Author response: 

Indeed there were three clusters with <10% of patients, with the smallest cluster consisting of 78 knees (5%), which is small but not negligible. Cluster size and trajectory group size was one of our considerations in deciding the number of clusters and trajectories, as mentioned in the Methods section [line 122-125 and 141-143]. Increasing the number of trajectories/clusters further may have resulted in us finding a lateral OA group but even when increasing the number of clusters up to 10 no clear lateral OA group emerged, and group sizes would likely have been much smaller, making results less applicable. As such indeed missing a lateral OA cluster may indeed not be an important limitation. We have included this in the Discussion section of the revised manuscript [line 313-317]. Failing to identify a lateral OA cluster is not considered a serious limitation, solely an unexpected observation. These are early OA patients, so perhaps there are simply not many patients with a clearly more severely affected lateral compartment. 

Reviewer 2

- Reviewer comment:

As this is a study in a supposedly “early” OA cohort, it is a great pity that the characterization at a structural level was limited to radiography, as using several MRI features (cartilage and meniscus damage, meniscal extrusion, bone marrow lesions, synovitis) would have been much more informative, with radiography missing all these tissue specific pathologies. Even JSW (and JSN) in radiography has been shown to not be specific to cartilage thickness, but is confounded by meniscus extrusion. The authors should specify this as a major limitation of the study. This being said, the current analysis is of course still very useful, since MRI is not currently used in the clinical routine of managing knee OA. 

- Author response:

We agree that the fact that structural evaluation was limited to radiography is a limitation. This was mentioned in the Discussion section, and we have now specified further that it is a major limitation in the Discussion section of the revised manuscript [line 350-351].

- Reviewer comment:

Yet, I think the authors need to provide more evidence that they are examining really phenotypes, and not just different stages of the disease. There appear to be patients with and without radiographic signs of TF and PF disease, and exhibiting different grades. How can the authors ensure that entering these variables, they are not just differentiating stages of the same disease (same path of progression), rather than different phenotypes (different paths). Similar considerations apply to the WOMAC scores.

- Author response:

This is a good point and we cannot be certain that we are not looking at different disease stages. However, there are several reasons why we think that we are not simply looking at different disease stages. First, there is no obvious chronologic line in the clusters (i.e. you cannot simply sort them from less to more advanced OA phase). Second, all patients have had their first complaints within 6 months of the first visit, meaning they are clinically all in the same phase of the disease at baseline, i.e. have the same starting point. Although this starting point is only based on clinical outcome, and not on radiographic features, it is very relevant in clinical practice. Further, the baseline KL grade and most tibiofemoral parameters were comparable and not significantly different between the found clusters/phenotypes (Table S4), so they likely did not start at very different structural stages either. Even if we derived trajectories represent different disease phases, groups would (at least partly) represent patients that progress more or less quickly through different phases, which may be regarded in itself a phenotype. We have added these considerations in the Discussion section of the revised manuscript [line 291-300].

ABSTRACT

- Reviewer comment:

Design: Please give a short hint (maybe in brackets) how early OA is defined

- Author response:

Early OA was defined as at or within 6 months of first visit to the general practitioner for these symptoms. We have added this to the Abstract of the revised manuscript [line 27-28].

- Reviewer comment:

How is the sentence: Combining….and radiographic (PF) parameters….

- Author response:

This sentence has been reworded to clarify our meaning [line 37-40].

- Reviewer comment:

It is unclear from my perspective whether the six clusters are only based on PF radiographic OA, and why TF radiographic parameters are not included.

- Author response:

This is because, when adding TF radiographic parameters, they were found to not significantly contribute to the clustering. We have clarified this in the Abstract of the revised manuscript [line 38-41].

- Reviewer comment:

When you say “gender”, I think this should be termed “sex”, since you are talking about biological and not social attributes. This comment also applies to the main text.

- Author response:

Yes this is correct, it should be termed ‘sex’, and have changed ‘gender’ to ‘sex’ throughout the entire manuscript. 

- Reviewer comment:

When including baseline parameters, does none of the radiographic features contributes to any of the clusters ?

- Author response:

‘Baseline characteristics’ would be a better way to describe this. We do not mean baseline parameters of WOMAC/TF/PF features, as these are already incorporated in the trajectories, and the subjects’ probabilities of belonging to each trajectory was used for clustering. The baseline characteristics that we used were: sex, menopausal status, age, BMI, biochemical markers in blood and urine, and pain-coping inventory (PCI) measures (described in the Methods section [line 136-139]). We have changed ‘baseline parameters’ to ‘baseline characteristics’ in the Abstract of the revised manuscript [line 41-43].

- Reviewer comment:

Conclusions: Although several…., they were shown…. Why are the first and second part of the sentence contradictions. The sentence is pretty general and could be a bit more specific.

- Author response:

These parts are indeed not clear contradictions. We have rewritten this sentence and specified it more as suggested in the Abstract of the revised manuscript [line 44-46].

- Reviewer comment:

Are really “gender” and “menopausal status” the main components making up 6-8 clusters (including men?), when phenotyping knee OA patients. From this sentence it appears that functional and radiographic (PF and TF) make not contribution. If this is so, please state this in the conclusion. Again, the conclusion should, in my view, be more concrete and more specific.

- Author response:

Indeed TF features did not contribute to the clustering, we did not make this clear in the abstract but have now done so in the Abstract of the revised manuscript in response to your earlier comment [line 40-41]. PF features did contribute, but their contribution was minor compared to sex, menopausal status, and WOMAC. We clarified this in the Abstract of the revised manuscript [line 41-43 and 46-48].

INTRODUCTION

- Reviewer comment:

P9, line 50: Please omit “already”

- Author response:

We have omitted ‘already’ from the revised manuscript [line 54].

- Reviewer comment:

P9, line 56: What is meant by “biochemical joint changes”. Please name the tissue you are referring to.

- Author response:

We have specified this in the revised manuscript by adding ‘in multiple tissues including bone, cartilage and synovium’ [line 61].

- Reviewer comment:

P9, line 59/60: … treatment….are available: Either “treatments” or “is available”

- Author response:

Thank you for noticing this, we have changed ‘treatment’ to ‘treatments in the revised manuscript [line 64].

- Reviewer comment:

P9, line 61: I suggest to say: disease “that consists of different phenotypes”

- Author response:

We have made the suggested change in the revised manuscript [line 66].

- Reviewer comment:

P9, line 64: suggest to add: “coalesce over time to a final common pathway. (delete “in patients”)

- Author response:

We have made the suggested change in the revised manuscript [line 69].

- Reviewer comment:

P10, line76: Please state in the introduction how you define “early” OA, as this term is very fashionable and it appears every researcher understands something different when using this term.

- Author response:

We have included this definition (first general practitioner visit ≤6 months before inclusion) in the Introduction of the revised manuscript as suggested [line 80-82]. 

- Reviewer comment:

P 10, line 78: Please omit “for the first time”

- Author response:

We have made the suggested change in the revised manuscript [line 84].

METHODS

- Reviewer comment: 

P10, line 84. As this study appears to focus on knee OA, but some patients also have hip OA, did you consider to include hip OA in your analysis and phenotypic characterization, as lower limb function may be impacted by the presence / absence of hip OA as well as knee OA.

- Author response:

Unfortunately there was no official diagnosis of whether patients had hip OA (or knee OA) or not, which makes it difficult to consistently control for hip OA. The fact that there is no significant difference in hip pain or stiffness between the clusters would indicate that hip OA likely did not have a significant influence on the clustering, but it could still have had an influence. For example, the WOMAC, although knee-specific, could have been influenced by patients having hip OA complaints as well. This consideration has been added to the Discussion section of the revised manuscript [line 360-362].

- Reviewer comment:

P10, lines 90/91: How can the patients have “severe complaints at baseline (most patients”, if this is a cohort of “early” disease?

- Author response:

More severe in this case is defined relative with this population of early OA patients. Indeed the criteria patients need to satisfy are still very mild: having 2 of pain, morning stiffness <30 min, crepitus and bony tenderness for the knee and/or 2 of pain, morning stiffness <60min pain on hip internal rotation or internal rotation <15°. We have now specified this in the revised manuscript [line 97-98].

- Reviewer comment:

P11, lines 95/96: Did you use WOMAC scores from one knee only (which one), or from both knees, as did the OAI. How did you account for contralateral knee (and hip) functional status in your analyses?

- Author response:

The WOMAC questionnaire was filled out on subject level instead of on knee level. The trajectory analyses were therefore also performed on subject level for the WOMAC, while they were performed on knee level for the radiographic features (see S1 Table). We have included this in the limitations [line 360-362].

- Reviewer comment:

P 11, line 98: Why did you use an absolute measure of JSW that varies substantially between patients of different sex and/or different body height, instead of using the common JSN grading (comparing the JSW with the less affected compartment and/or contralateral knee), as a real measure of pathology.

- Author response:

We used JSW because it is a continuous measure and as such more sensitive to change, allowing us to identify trajectories more specifically. For PF features we used JSN, as continuous JSW was not available. 

- Reviewer comment:

P 11, line 98: Why did you use osteophyte “area” rather than the common semi-quantitative grading system?

- Author response:

Similar to the JSW, we used a continuous measure for osteophyte size as it is more sensitive to change, allowing us to identify trajectories more specifically. For PF features there was no continuous measure for osteophytes available, so we used semi-quantitative scores. 

- Reviewer comment:

P 11, line 99/100: Why did you use a quantitative system for the TF joint, and a semi-quantitative one for the PF joint? As you appear to find a greater contribution of the PF than TF joint to the clusters, may this be due to PF radiographic grades being used semi-quantitatively (as a sign of pathology), whereas the TF joint is evaluated quantitatively, where absolute measures are not really a sign of radiographic pathology, but are strongly confounded by inter-subject variation.

- Author response:

For the clustering, only the trajectory probabilities were included, so in this sense the direct influence of parameters being measured continuously or semi-quantitatively is limited. However, it may indeed have influenced the identification of trajectories themselves. For example part of the TF trajectories could be inter-subject variation, meaning the found trajectories are not a full reflection of the actual progression, or trajectories might have been missed using grading scores. This is a good point and we have added this to the Discussion of the revised manuscript [line 279-283].

- Reviewer comment:

P 11, line 106: Thereafter…. Is this then referring only to the group included. This is a bit confusing.

- Author response:

Indeed this is only referring to the included records, we clarified this in the revised manuscript [line 121]. 

- Reviewer comment:

P 11, line 111: domain “experts”. Is this word used correctly here. May be due to my unfamiliarity with this particular method.

- Author response:

‘Experts’ here merely refers to graphical evaluation and exploration of population differences in clusters by subject matter experts, it is not a statistical term. We understand the confusion and have rephrased this in the revised manuscript [line 124].

- Reviewer comment:

P 12, lines 115-117: Can you give the non-specialist reader a sense why the non-functional data is possibly relevant to define phenotypes (and not the functional analysis)?

- Author response:

Both the non-functional and functional data are important to define phenotypes and both were used. The only difference is that it is not possible to use both types of data in one analysis, so we first performed the functional data analysis and used the probabilities of belonging to the identified trajectories as a way to use both types of data in the same analysis for defining subgroups. We have clarified this (also based on the comment below on explaining better what the various methods do conceptionally) in the revised manuscript [line 128-134].

- Reviewer comment:

P12, lines 120-123: It appears that the PF and TP radiographic features were not included at this level; why not?

- Author response:

PF and TF radiographic features were included at this level. We should have clarified this better and did so in the revised manuscript [line 135].

- Reviewer comment:

P12, lines 115 ff: Could you try to give the reader a sense of what these various methods do conceptionally, as not every reader may be familiar with theses statistical methods.

- Author response:

Functional data analysis allows for division/clustering of subjects based on the course over time of multiple separate characteristics. To define possibly relevant phenotypes, patients should be clustered based on both their characteristics (non-functional data) and their (clinical and structural) progression over time (functional data), but this is not (directly) possible in a single analysis. To combine the functional data and non-functional data we firsts ‘reduced’ the functional data to a probability of membership of a subject/knee to belong to a specific trajectory group. Expressed in this way it can be regarded a characteristic of a patient/knee and can be combined with the non-functional data (i.e. subject characteristics possibly relevant to define phenotypes) in a hierarchical cluster analysis to define subgroups. We have included this explanation in the revised manuscript [line 106-108 and 128-134].

RESULTS

- Reviewer comment:

P13, lines 138-141: Why do you think specifically the TF JSW and PF JSN differ between included and excluded participants, but none of the other parameters?

- Author response:

We think this is because all patients that were not part of the more ‘severe’ group were not followed yearly, and thus missed 3 or more visits and were excluded. As such, it makes sense that the excluded patients have (slightly) less severe OA, although this was not the case when comparing WOMAC. We highlight this in the Discussion section of the revised manuscript [line 345-350]. 

- Reviewer comment:

P13, lines 141 ff: it was decided…. based on which criteria

- Author response:

This was because no clear patterns could be obtained using all features in combined analyses. We reworded this part in the revised manuscript to clarify [line 157-161].

- Reviewer comment:

P13, lines 149 ff: Why was the WOMAC score not determined at the knee level (as was done in OAI). Which knee was selected for the analysis regarding WOMAC in each patient? How do the authors expect the contralateral knee to confound the scores?

- Author response:

Unfortunately there was no WOMAC on knee level available in the CHECK cohort. The WOMAC trajectory analysis was done on patient level and not on knee level for this reason, but in the cluster analysis this could indeed have confounded the scores. We have included this in the limitations in the Discussion section of the revised manuscript [line 360-362].

- Reviewer comment:

P13, lines 160 ff: How did you define medial knee OA in the absence of radiographic JSN scores?

- Author response:

We meant that the fact that all found trajectories shows decreasing medial JSW and increasing lateral JSW suggests patients are primarily affected on the medial side, since it could be expected that the more affected compartment decreases over a period of 10 years (and the less affected compartment might increase as a result of wedging). However we cannot say for certain, since we did not officially determine the most affected compartment, and have added this in the revised manuscript [line 178].

- Reviewer comment:

P14, line 184: Adding WOMAC…. Adding to what ?

- Author response:

‘Adding’ was not chosen well, we meant to say that we performed Hierarchical Cluster Analysis with the WOMAC trajectories. We have reworded this sentence in the revised manuscript [line 201-202].

- Reviewer comment:

P15, lines 190 ff: I wonder whether the lack of relevance of TF OA is due to the lack of use of JSN scores, as JSW is not a measure of pathology in a cross sectional context?

- Author response:

Yes this is a good point in line with an earlier comment on the difference between semi-quantitative and continuous measures. We have added this to the Discussion of the revised manuscript [line 279-283].

- Reviewer comment:

Can you provide results on what the JSW change was over 10 years in the medial and in the lateral compartment, respectively?

- Author response:

The JSW change over time in the medial and lateral compartment can be seen in Fig 1. (In numbers, the lateral compartment showed an increase of 1.15 mm (95%CI 1.07-1.23) and the medial compartment a decrease of 0.45 mm (95%CI -0.49 – -0.41)).

- Reviewer comment:

P15, line 198 : Please do not say “unfortunate”, the presentation of results should be neural

- Author response:

We removed ‘unfortunately’ from the results of the revised manuscript [line 216].

- Reviewer comment:

P15, lines 200 ff: When sex and menopause were added (how did you classify men in this context), did really all the WOMAC and radiographic PF OA data became irrelevant (Therefore we decided to remove the other characteristics……)

- Author response:

Sex and menopause were one variable categorized as men, premenopausal women, postmenopausal women (see line 218). WOMAC and PF OA data did not become irrelevant when adding sex and menopause, we meant that the other added baseline characteristics became irrelevant. We have clarified this in the revised manuscript [line 219].

- Reviewer comment:

P15, lines 206ff: In my view, these phenotypes appear to represent different stages of the disease.

- Author response:

This is a good point and we cannot be certain that we are not looking at different disease stages. However, there are several reasons why we think that we are not simply looking at different disease stages. First, there is no obvious chronologic line in the clusters (i.e. you cannot simply sort them from less to more advanced OA phase). Second, all patients have had their first complaints within 6 months of the first visit, meaning they are clinically all in the same phase of the disease at baseline, i.e. have the same starting point. Although this starting point is only based on clinical outcome, and not on radiographic features, it is very relevant in clinical practice. Further, the baseline KL grade and most tibiofemoral parameters were comparable and not significantly different between the found clusters/phenotypes (Table S4), so they likely did not start at very different structural stages either. Even if we derived trajectories represent different disease phases, groups would (at least partly) represent patients that progress more or less quickly through different phases, which may be regarded in itself a phenotype. We have added these considerations in the Discussion section of the revised manuscript [line 291-300].

- Reviewer comment:

P 15/16: The listing of the different clusters in the text is a bit difficult to follow. Maybe these could be put into Tables, cross-tabling the different features. This will also display visually which combinations were not in the set of selected phenotypes.

- Author response:

We understand that listing the different clusters is a bit difficult to follow. A Table cross-tabling all different features and their contribution would be difficult to execute, since there are no clear cut-off points when features do and do not contribute and the grouping/naming was performed on clinical interpretation as well. However, we have clarified the legends of Fig 4 and Fig 5 to more clearly present the different clusters, and hope this makes the listing of the clusters easier to follow [line 236-240 and 253-257]. 

DISCUSSION:

- Reviewer comment:

P 16, lines 238 ff.: It would be beneficial if the structure of presentation of results could differ more clearly between cross sectional and longitudinal (change) distributions of the various features.

- Author response:

In the trajectory analyses we never look at cross-sectional distributions. We realize that the word ‘absolute’ here is unclear, we meant that trajectories could be classified based on absolute levels over time (e.g. constant high level or low level over 10 years) and the course over time. We have clarified this in the Discussion of the revised manuscript [line 262].

- Reviewer comment:

P 17: When talking about TF and PF OA here, please add “radiographic” to the description.

- Author response:

We have added ‘radiographic’ to ‘TF OA characteristics’ and ‘PF OA characteristics’ the first time they are mentioned in the Discussion section [line 273-275].

- Reviewer comment:

P 18, lines 268/269: How many of your participants had medial vs. lateral TF radiographic OA?

- Author response:

We do not know, as this was not officially determined. We have added this to the discussion of the revised manuscript [line 313-315].

- Reviewer comment:

P 19, line 291: Why do the authors think the results are so discordant between studies, given that they have examined a relatively large cohort.

- Author response:

The one clear difference between our study and previous studies is that previous studies showed significant differences based on BMI and we did not. We think this could be because the other studies clustered based only on pain, while we looked at other factors as well, and BMI is known to be linked to pain. We have included this in the Discussion section [line 338-340].

- Reviewer comment:

P 19, line 293: Instead or “some missing data was present”, please say “ some data was missing”.

- Author response:

We have made the suggested change in the revised manuscript [line 342].

- Reviewer comment:

P 19, lines 308 ff: Shouldn’t the Osteoarthritis Initiative (OAI) data be used for validation?

- Author response:

Yes it is a good suggestion to add this and we have done so in the revised manuscript [line 363-364].

- Reviewer comment:

P 20, conclusions: There is more text about what was done than about what was found. Please focus more on concrete and specific findings in the conclusion section (also in the abstract)

- Author response:

We have changed the conclusion of the abstract [line 44-49] and the manuscript [line 373-382] to focus more on our findings. 

- Reviewer comment:

P 20, conclusions: Please do not state “for the first time”, since the study is one in line of such analyses, and of course every study has its peculiarities, but these to not necessarily qualify for stating “for the first time”, if similar approaches have been taken before.

- Author response:

We removed ‘for the first time’ from the revised manuscript as suggested [line 381].

---

## [Decision Letter · Decision Letter 1]

15 Mar 2023

Identifying multivariate disease trajectories and potential phenotypes of early knee osteoarthritis in the CHECK cohort

PONE-D-22-19232R1

Dear Dr. Jansen,

We’re pleased to inform you that your manuscript has been judged scientifically suitable for publication and will be formally accepted for publication once it meets all outstanding technical requirements.

Kind regards,

Aqeel M Alenazi

Academic Editor

PLOS ONE

Additional Editor Comments (optional):

Reviewers' comments:

Reviewer's Responses to Questions

**Comments to the Author**

1. If the authors have adequately addressed your comments raised in a previous round of review and you feel that this manuscript is now acceptable for publication, you may indicate that here to bypass the “Comments to the Author” section, enter your conflict of interest statement in the “Confidential to Editor” section, and submit your "Accept" recommendation.

Reviewer #2: All comments have been addressed

2. Is the manuscript technically sound, and do the data support the conclusions?

Reviewer #2: Yes

3. Has the statistical analysis been performed appropriately and rigorously? 

Reviewer #2: I Don't Know

4. Have the authors made all data underlying the findings in their manuscript fully available?

Reviewer #2: Yes

5. Is the manuscript presented in an intelligible fashion and written in standard English?

Reviewer #2: Yes

6. Review Comments to the Author

Reviewer #2: Thank you for the detailed response and author action to the comments made.

Thank you for the detailed response and author action to the comments made.

7. PLOS authors have the option to publish the peer review history of their article (what does this mean?). If published, this will include your full peer review and any attached files.

Reviewer #2: No

---

## [Editor Report · Acceptance letter]

24 Mar 2023

PONE-D-22-19232R1 

Identifying multivariate disease trajectories and potential phenotypes of early knee osteoarthritis in the CHECK cohort 

Dear Dr. Jansen:

I'm pleased to inform you that your manuscript has been deemed suitable for publication in PLOS ONE. Congratulations! Your manuscript is now with our production department. 

Kind regards, 

on behalf of

Dr. Aqeel M Alenazi 

Academic Editor

PLOS ONE